# Immunological Signatures in Blood and Urine in 80 Individuals Hospitalized during the Initial Phase of COVID-19 Pandemic with Quantified Nicotine Exposure

**DOI:** 10.3390/ijms25073714

**Published:** 2024-03-27

**Authors:** Krzysztof Laudanski, Mohamed A. Mahmoud, Ahmed Sayed Ahmed, Kaitlin Susztak, Amal Mathew, James Chen

**Affiliations:** 1Department of Anesthesiology and Perioperative Care, Mayo Clinic, Rochester, MN 55902, USA; chen.james@mayo.edu; 2Department of Pulmonary and Critical Care, Mayo Clinic, Rochester, MN 55902, USA; mahmoud.mohamed@mayo.edu (M.A.M.); ahmed.ahmed5@mayo.edu (A.S.A.); 3Department of Nephrology, University of Pennsylvania, Philadelphia, PA 19146, USA; ksusztak@pennmedicine.upenn.edu; 4School of Biomedical Engineering, Science and Health Systems, Drexel University, Philadelphia, PA 19104, USA; agm76@drexel.edu

**Keywords:** SARS-CoV-2, COVID-19, nicotine exposure, cotinine, immunological response, spike protein, immunoglobulin, cytokines, blood, urine

## Abstract

This research analyzes immunological response patterns to SARS-CoV-2 infection in blood and urine in individuals with serum cotinine-confirmed exposure to nicotine. Samples of blood and urine were obtained from a total of 80 patients admitted to hospital within 24 h of admission (t_adm_), 48 h later (t_48h_), and 7 days later (t_7d_) if patients remained hospitalized or at discharge. Serum cotinine above 3.75 ng/mL was deemed as biologically significant exposure to nicotine. Viral load was measured with serum SARS-CoV-2 S-spike protein. Titer of IgG, IgA, and IgM against S- and N-protein assessed specific antiviral responses. Cellular destruction was measured by high mobility group box protein-1 (HMGB-1) serum levels and heat shock protein 60 (Hsp-60). Serum interleukin 6 (IL-6), and ferritin gauged non-specific inflammation. The immunological profile was assessed with O-link. Serum titers of IgA were lower at t_adm_ in smokers vs. nonsmokers (*p* = 0.0397). IgM at t_48h_ was lower in cotinine-positive individuals (*p* = 0.0188). IgG did not differ between cotinine-positive and negative individuals. HMGB-1 at admission was elevated in cotinine positive individuals. Patients with positive cotinine did not exhibit increased markers of non-specific inflammation and tissue destruction. The blood immunological profile had distinctive differences at admission (MIC A/B↓), 48 h (CCL19↓, MCP-3↓, CD28↑, CD8↓, IFNγ↓, IL-12↓, GZNB↓, MIC A/B↓) or 7 days (CD28↓) in the cotinine-positive group. The urine immunological profile showed a profile with minimal overlap with blood as the following markers being affected at t_adm_ (CCL20↑, CXCL5↑, CD8↑, IL-12↑, MIC A/B↑, GZNH↑, TNFRS14↑), t_48h_ (CCL20↓, TRAIL↓) and t_7d_ (EGF↑, ADA↑) in patients with a cotinine-positive test. Here, we showed a distinctive immunological profile in hospitalized COVID-19 patients with confirmed exposure to nicotine.

## 1. Introduction

Smoking severely impairs lung performance, with some patients developing chronic obstructive pulmonary disease (COPD) after prolonged exposure [1]. Short-term cigarette smoke exposure results in an imbalance between free radicals and antioxidants, accelerated cellular senescence, and diminished levels of anti-protease inhibitors while being immunomodulatory in several ways [2,3,4]. The long-term effect of exposure to nicotine and tar products impacts the immune system and lungs via cellular reprogramming. Smoke particles irritate the lungs, with autophagy and necroptosis significantly contributing to deteriorated lung function [4,5,6,7]. Additionally, increased levels of nicotine and tobacco-related products impact immune function in a variety of ways, including direct interaction with leukocytes as well as hormonal and nervous pathways [8,9,10]. Some of these changes are reversible, while others are related to cellular reprogramming or result from direct tissue damage [11,12]. Eventually, nicotine exposure accelerates organ failure by increasing their susceptibility to stressors. The net effects of nicotine exposure are detrimental, resulting in increased susceptibility to infection and less favorable outcomes in the case of pathogen challenge [13,14,15].

The outcome of the SARS-CoV-2 infection is determined by individual susceptibility, viral load, and immune system response [16,17,18]. In general, during the initial wave of infection, viral pathogens enter pneumocytes aided by angiotensin-converting enzyme receptors [18,19]. Unsurprisingly, patients with concomitant COVID-19 and aberrant renin–angiotensin–system (RAS) (diabetes, hypertension, kidney disease) have increased mortality, which only partially can be accounted for by pre-existing morbidity [19,20]. Subsequent unopposed proliferation of the virus results in necrosis of pneumocytes with subsequent release of the danger-associated molecular patterns (DAMPS) and viral particles [18,21]. The ensuing immune response is often characterized as a cytokine storm. The natural history of immune system response is deemed as one of the primary determinants of COVID-19 outcomes [22,23,24,25,26,27,28]. However, the immune response is far from uniform in affected individuals in general and during COVID-19 [16]. This heterogeneity of initial immune responses may account for variance in COVID-19 mortality and morbidity [16,17,24,26,29]. 

Smoking and tobacco ingestion affect several steps in SARS-CoV-2 infection [30,31,32,33]. In most general terms, nicotine exposure empowers the virus to be more pathogenic while immunomodulatory properties of nicotine tamper with immune response [6,10,11,12,30,32]. Unimpeded virus proliferation due to immuno-incompetency is a reason for increased viral pathogenicity [34]. Clinical data demonstrated that nicotine increases SARS-CoV-2 entry into pneumocytes, as predicted early in the pandemic [31,33,35,36,37]. Also, cellular aging secondary to prolonged smoking exposure may contribute to viral proliferation due to the senescent or suppressed immune system [3,4,9,10,30,38]. Nicotine-induced activity of RAS may modulate the pathogenicity of the virus directly [31,33]. Finally, smokers have a high incidence of hypertension resulting in a more common intake of ACE inhibitors. The net effect is exaggerated significant lysis, NETosis, ferroptosis, and apoptosis of the pneumocytes due to the higher exposure to viral load [5,18,21,23]. This may trigger a more severe cytokine storm in individuals exposed to nicotine. Conversely, nicotine may be immunomodulatory via selectively effective cytokine milieu and cholinergic activity, resulting in a much more moderated cytokine storm [6,30,32]. These complex pro-viral and immuno-aberrant conditions are juxtaposed on significantly reduced reserve to cope with respiratory infection caused by SARS-CoV-2 or other pathogens in the case of individuals with prior exposure to smoking [1,13,14,15]. The net effect of nicotine on COVID-19 remains a hot debate, yet no data assessing nicotine intake, viral load, and immune response in the context of clinical outcomes have been collected [34,39,40]. 

Data from over 30 studies analyzing the effect of smoking on COVID-19 outcomes demonstrated a complex effect on mortality [39]. This mixed outcome results from complex interactions between nicotine, immunity, and the respiratory system. Most studies showed an increased mortality, while few demonstrated no effect. In one study, smoking seemed to have a protective effect. Only one study looked into the dose effect of smoking with a highly positive correlation. In most of the studies, COVID-19 was more severe. However, all these conclusions are severely limited by several covariables. Incidence of smoking was self-reported or drawn from electronic health records (EHR) in virtually all studies. Reliance on EHR is subjected to numerous biases, including the fact that the most completed and accurate medical records are usually biased towards healthier people, and the collection of the data regarding smoking is often difficult to trace and validate as the self-reported data are frequently subjective [41,42]. Furthermore, the definition of smoking varied with particular heterogeneity in characterizing former smokers. The quantification of smoking in individuals reporting utilization of tobacco in the past is particularly difficult. Outcome definitions varied greatly as well across the studies.

The difficulty in predicting the interaction between tobacco intake and the outcome of COVID-19 encounters additional problems as the characterization of the immune system response to nicotine intake is also complex. The immunological response is often analyzed in terms of initiating pathogen versus pathogen-specific response versus non-specific response instead of providing a holistic overview of the host’s responses. Most immune system measurements are carried out using blood as the specimen [23,24,25,28,29]. However, invasive techniques have to be used to obtain the sample. Consequently, increasing stress is placed on analyzing other sources of immunological information, including urine, sweat, stool, saliva, and others [35,43,44]. Urine can reflect the immunological performance in general as well. Additionally, it provides a window into the immune system response in the kidneys.

In this manuscript, we decided to analyze the immunological response to SARS-CoV-2 infection in individuals with cotinine-tested exposure to tobacco [42,45,46,47,48]. We analyzed viral load, immunological response, and serum levels of danger-associated proteins in nicotine-positive vs. nicotine-negative individuals. We presumed that individuals with positive cotinine testing would have a more pronounced viral load and elevated markers of tissue destruction. We hypothesize that smoking history will affect the longitudinal immunological response, potentially explaining less favorable clinical outcomes in smokers affected by COVID-19.

## 2. Results

### 2.1. Characteristics of the Sample and Prevalence of EHR-Reported and Serum-Tested Cotinine Levels

A total of 80 patients were enrolled in the study and had serum cotinine tested. A total of 47.5% had no history of tobacco exposure during their entire life, 12.5% self-reported current tobacco use, and 40% admitted past nicotine use. Demographic data on one patient were missing. 

When the serum level of cotinine was tested, 81.01% of enrolled individuals had cotinine levels below the threshold, while 18.99% had detectable cotinine levels. When we compared patients divided by the cotinine threshold level, a significant difference was apparent (M_eCotininePOS_ = 37.5 [7.97; 100.2] vs. M_eCotininNEG_ = 1.7 [0.46; 1.55], U[79] = 5.99; *p* < 0.0001) with some users having exceptionally high levels (Figure 1).

When we analyzed the group of patients with positive tests for cotinine, 43.75% were negative for smoking history from electronic health records, 42.5% reported a past medical history of smoking, and 13.75% were active smokers. Nobody reported using vaping as a source of nicotine.

### 2.2. Effect of Smoking on Demographical Variables of the Patients

The demographic and clinical characteristics of the patients enrolled in the study (*n* = 80) are presented in Table 1. They tend to be more chronically sick judging from slightly higher CCI scores, but the difference was not statistically significant [CCI (X; SD) CCI_CotininePOS_ = 4.5 ± 3.09 vs. CCI_CotinineNEG_ = 3.6 ± 2.94; *p* = 0.3]. No difference in positive cotinine tests was detected when demographical and clinical characteristics were considered including age. The only correlation we found is that age correlated with length of stay (*r^2^* = 0.26; *p* = 0.025) or length of mechanical ventilation (*r^2^* = 0.24; *p* = 0.038) but no statistically significant differences were seen between cotinine-positive and negative subjects.

### 2.3. Viral Load, Immunoglobulin Titers, Level of Tissue Necrosis, and Non-Specific Inflammatory Markers Assessed in Patients with Cotinine Levels in SARS-CoV-2 Infection

As judged by serum the S-protein level between admission to 7 days after infection, the viral load was similar between cotinine-positive and negative patients (Figure 2A).

The level of IgM against the S and N proteins was lower at t_48h_, but the initial and delayed measurements were similar between cotinine-positive and negative subjects (Figure 2B). The levels of IgA at admission were lower in patients who tested positive for cotinine (Figure 2C). IgG was no different between cotinine-positive and negative (Figure 2D).

HMGB-1 was elevated at the admission when a one-sided hypothesis was considered but serum heat shock protein 60 was not significantly different at that time (Figure 3A,B). Markers of non-specific inflammation (ferritin, IL-6) were not different between these two groups of patients at any sampling time (Appendix A).

### 2.4. Acute Immunological Blood of Blood Serum in Cotinine-Positive Individuals at Admission

The blood immunological profile demonstrated several differences at the sampling point in 10 markers out of 96 total in the OLINK panel (Figure 4). Chemokine ligand 19 (CCL19) and major chemokine protein 3 (MCP-3) were downregulated in patients with positive cotinine tests at _48h_ (Figure 4). Cluster of Differentiation 28 (CD28) in the serum exhibited biphasic evolution with elevation at t_48h_ and downregulation at t_7d_. Cluster of differentiation 8A (CD8A) in the serum was depressed at 48 h in cotinine-positive individuals. Concomitantly, interferon γ (IFNγ) and IL-12 were depressed at t_48h_ in the same individuals. The serum granzyme B (GZNB) level was elevated at t_48h_ if the cotinine test was positive. The serum level of MHC class I polypeptide-related sequence A (MIC A/B) was downregulated at both t_adm_ and t_48h_ in cotinine-positive patients. Several of these differences demonstrated *d*-Cohen in the moderate impact range (Appendix A). If a one-sided hypothesis is considered, we found several additional markers (CXCL10_t48_, GZMA_t48_, IFNγ_tadm_, IL-10_t48_, KLRD_t48_, MUC-16_tadm_, TNFα_tadm_, TNFRSF9_t7d_) (data not shown) but we did not include a specific hypothesis about their behavior during COVID-19 as one could not be formulated.

### 2.5. Urine Profile during COVID-19

The urine immunological profile demonstrated several differences at the sampling point in 9 markers out of 96 total in the OLINK panel (Figure 5). Chemokine ligand 20 (CCL20) demonstrated an increased urine level at t_adm_ and decreased at t_48h_ in patients with positive tests. C-X-C motif chemokine 5 (CXCL5) was consistently elevated at t_adm_ if cotinine testing was positive at admission. Tumor necrosis factor-related apoptosis-inducing ligament (TRAIL) at t_48h_ and epidermal growth factor (EGF) at t_7d_ were elevated similar way in individuals testing positive for tobacco. IL-12 and CD8A were altered in smokers at t_adm_. Tumor Necrosis Factor Ligand Superfamily Member 14 (TNSFS14), granzyme H (GNZNH), and MIC A/U were significantly altered if cotinine was positive at admission. Finally, the urine level of amino deaminase (ADA) was increased at t_7d_ in case of a positive cotinine test. Several of these differences demonstrated *d*-Cohen statistics in the moderate impacts range (Appendix A). If a one-sided hypothesis was considered, we found several additional markers (ARG_tadm_, CXCL10 _t48h_, CXCL11_t48h_, DCN_t48h_, GZMB_tadm_, HO1_t48h,_ IL-18_t48h_, KIRDL1_tadm_, MCP-4_t48h_, MMP12_t7d_, MMP7_t72h_, TIE_tadm_) but we did not include a specific hypothesis about their behavior during COVID-19 could not be formulated.

## 3. Discussion

We found that a positive cotinine test was linked to subdued initial IgM and IgA titer early during illness in hospitalized patients while triggering specific blood (MIC A/B↓, CCL19↓, MCP-3↓, CD8A↓, IFNγ↓, IL-12↓, granzyme B↓) and urine (CCL20↑, CXCL5↑, CD8A↑, IL-12↑, MIC A/B↑, granzyme H↑, TNSRFS14↑, CCL20↓, TRAIL↓, EGF↑, ADA↑, MIC16↑) signatures. These differences were time dependent, with some markers having differences as discrete sampling times, while only one exhibited biphasic dynamic (CD28↑=>↓) in the cotinine-positive group. The blood profile indicates exhaustion of T cell and monocyte activation, with a possible predilection for the cytotoxic type [2,16,17,22,25,49,50]. T-cell hyperactivation was linked to less favorable outcomes of COVID-19, but none of the prior studies examine smoking as a potential confounder [18,26,29,35,51]. However, our data may suggest that nicotine abuse, at minimum, is a significant confounder in response to COVID-19. The pattern of immune activation may provide insight into the etiology of unfavorable COVID-19 pathology. Monocyte activation and translocation into lung parenchyma are linked to lung damage [50]. Similarly, IFNγ and IL-12 production abnormalities are linked to more severe COVID-19 outcomes as both cytokines are critical for effective antiviral response [51]. It is to be determined if low interferon can correlate with the emergence of fibrosis post-COVID-19, but our data showed normalization at day seven [52]. Elevated granzyme B and H levels are part of the initial granulocyte response during non-specific immune system activation [53]. The immunological signatures described in this manuscript are complex and do not represent a simply interpretation. This is consistent with the current understanding of COVID-19 as an illness of immune dysregulation, not hyperactivation [54]. Some of the observations are consistent with the overall effect of nicotine on the immune system [30,31,32,40]. However, the interpretation may be more complex as nicotine affects several aspects of SARS-CoV-2 interactions with the host [19,31,33,34].

One of the most exciting findings of our study is the depleted titer of IgA and IgM during the initial phase of infection. Prior data showed that smoking history can affect immunoglobulin levels with increasing titer of IgM and IgA while depressing IgG [55,56,57,58]. Furthermore, the different IgG subclass changes were very complex [59,60]. Stopping smoking resulted in a rebound of IgA [60]. Here, we showed that specific levels of immunoglobulins against specific antigens SARS-CoV-2 are altered during the initial phases of infection. These alterations in the initial specific response to COVID-19 may have a profound impact as the emergence of acquired immunity is critical to suppressing non-specific cytokine storms. Considering that COVID-19 is related to abnormalities in cytokines critical to immunoglobulin production, it is not entirely surprising that we see specific differences [23]. The lack of the effect of smoking on viral load or tissue necrosis suggests that immune signatures are modified by smoking itself, not by affecting pathogen load or innate immune response.

Finding several differences between urine and blood compartments in immunological profiling is exciting and novel. Urine sampling reflects regional tissue responses, while blood represents a gestalt of the immunological response. Nicotine affects organs more than blood as it accumulates in them. This is one of the reasons why COPD is the primary presentation of prolonged nicotine exposure. Here, we see a profound effect of nicotine intake, which may reflect the profound effect of nicotine on the kidney as the end-organ processing several products of nicotine ingestion.

This study has some advantageous points. We conduct confirmatory tests for cotinine in serum. This test accounts for active and passive nicotine product exposure to different forms of nicotine. In contrast, most of the studies used a medical history of interviews to determine exposure to tobacco. This assessment of smoking is subjected to reporting bias and does not account for the quantification of exposure levels [41,47]. From our data, the congruency of cotinine serum levels of EHR-based data was low, confirming prior observation [41,42,45,47]. Consequently, we relied on cotinine levels to measure nicotine exposure [46,48]. However, even with this approach, we are unable to assess the duration of smoking, a significant factor determining harmful levels of exposure to nicotine. We are also unable to ascertain the source of nicotine. Nicotine can be inhaled, ingested, or absorbed via the transdermal route. It could be actively or passively consumed. This is an important factor as utilization of cigarette smoking results in the acquisition of additional toxins affecting immune system activation regardless of the source. However, additives to nicotine delivery vehicles may have a significant impact on toxicity. In the most recent iteration, nicotine can be inhaled with e-cigarettes. Their preparation often contains additional additives with complex effects on host immunity and homeostasis [61,62]. We utilized the protein level to assess viral load instead of RNA load. In contrast to PCR-based methods, the protein level is not subjected to residual SARS-CoV-2 ribonucleic acid levels, which may persist for a protracted period. We found that smoking affects levels of immunoglobulin, as previously published [11,56,59,60]. This is a potential mechanism for how smoking affects the long-term outcome of COVID-19 as the evolution of acquired immunity is pivotal for tampering with uncontrolled cytokine responses [13,22,26,27,29]. However, the report of cytokine storms in COVID-19 has to be taken with a grain of salt as immunosuppression has also been reported [28]. The OLINK profile resembled more cytokine suppression and T cell exhaustion, especially a couple of days after admission [19,23,24,63]. To conclude, the effect of nicotine intake on the outcome of sepsis may be diverse [32].

This study has some limitations specific to its methodology. We only enrolled patients with severe symptoms to be hospitalized. Consequently, we do not know how smoking impacts COVID-19-triggered cytokine storms in patients exposed to nicotine but who are asymptomatic [30,32,33]. Presumably, those patients are able to fend off infection effectively despite the impact of nicotine ingestion. There is a significant bias on long-term surveillance as the patients who are more likely to be sick are captured due to the higher utilization of healthcare services. Our cohort is over-representing the African American population. This is consistent with observations by others during the first wave of the pandemic. However, race has a profound effect on immunoglobulin, co-existing illnesses, and duration of treatment. This study did not account for some additional variables co-existing with a history of tobacco use (comorbidities, advancement of COPD, concomitant use of steroids, etc.) that future studies should explore [1,14,15,41]. The sample size of our study cohort is small enough to compute meaningful comparisons for several markers as the heterogeneity of cytokine response is significant. The frequency of COPD between cotinine-positive versus negative users was non-significant while expected for COVID-19-stricken patients. However, we only measured cotinine levels in a short window of time. Consequently, some of our patients could develop COPD and, meanwhile, quit smoking [1]. The cotinine concentration varied considerably, yet our sample size was insufficient to ascertain if the observed effect of nicotine correlated with the serum level of this compound. It is also very likely that cotinine exposure effects are non-linear in some physiological aspects. On the other hand, deterioration of pulmonary function likely has a cumulative effect, whereas acute and chronic exposure to nicotine has a different impact on the host’s immune system [40]. Finally, the source of nicotine varies in terms of impact on pulmonary and immune system function. Vaping and smoking cigarettes have added harm to pulmonary function, while tobacco replacement using a transdermal delivery system affects the pulmonary system less [46,48,61,62]. Dissecting these interactions will require a robust study involving a much larger patient sample. The inherited difficulty is the robustness of the self-reported data, as smokers often underestimate the amount of cigarettes and the duration of their dependence [1]. We did not test serum levels of the angiotensin-converting enzyme as it rapidly declines during acute illness to recover during convalescence [19,31,33,40].

This research is exploratory in generating an idea that the immunological response to COVID-19 can be profiled. We found that some of our research overlapped with others [16,17,43,56,60]. Due to the abundance of caution, we did not conduct extensive correlational analysis, which is common in the works of others. Here, we set out to demonstrate the difference between smokers and non-smokers in general and purposefully conduct our analysis in a very conservative way. We believe the next step is to include a more diverse cohort and conduct potential animal studies.

## 4. Materials and Methods

### 4.1. Ethical Concerns

The Institutional Review Board of the Hospital of the University of Pennsylvania approved this study (#813913). The principles of the Declaration of Helsinki were followed throughout this study.

### 4.2. Study Cohort and Clinical Data Extraction

Patients admitted to the hospital from March 2020 to December 2020 with PCR-confirmed diagnoses of SARS-CoV-2 were approached and recruited for the study. The demographic and clinical data were collected using electronic health records (EHR). 

Patients self-determined their race and ethnicity. The Acute Physiology and Chronic Health Evaluation II (APACHE II) score was calculated within 1 hour (APACHE_1h_) and 24 h after admission (APACHE_24h_) [64]. The burden of chronic disease was calculated using the Charlson’s Comorbidity Index (CCI) [65]. Utilization of remdesivir, convalescent plasma, and steroids was extracted from EHR. Except for the latter, the treatments were highly protocolized per hospital policy and according to the FDA recommendations for the given treatment. Steroid treatment was defined as using intravenous or oral glucocorticoid steroid compounds to treat COVID-19 pneumonia, per the healthcare provider’s notes.

### 4.3. Study Procedure and Sample Collection

The potential study subject was approached shortly (<24 h) after admission to the hospital (t_adm_). The first blood samples were collected upon consent, followed by another at 48 h (t_48h_), and 7 days (t_7d_).

Blood was collected in BD Vacutainer™ tubes (BD, Franklin Lakes, NJ, USA) with heparin as an anticoagulant. The sample was centrifuged at 2000× *g* for 10 min at 4 °C to separate the serum. The serum was subsequently aliquoted and stored at −80 °C. Per protocol, all samples were inactivated with (5%) Triton X-100 (ChemCruz, Dallas, TX, USA).

Urine was collected using standardized hospital equipment, aliquoted, and stored at −80 °C. Per protocol, all samples were inactivated with (5%) Triton X-100 (ChemCruz, Dallas, TX, USA). 

### 4.4. Assessment of the Viral Load, the Release of Danger-Associated Molecular Patterns, and Non-Specific Immune Response

The level of S-protein was measured using commercial kits (RayBiotech, Stanford, CA, USA). The IgM, IgG, and IgA levels against proteins S and N were measured using commercially available kits (RayBiotech, Stanford, CA, USA) per manufacturer information [35]. In addition, commercial enzyme-linked immunoassays were used to measure heat shock protein-60 (Hsp-60) (Thermo Fisher Scientific, Waltham, MA, USA) and high mobility box-1 protein (HMGB-1) (Aviva, Auburn, MA, USA) to assess the level of cellular destruction and release of DAMPs [66,67].

### 4.5. Immune Markers Testing

Non-specific inflammatory (ferritin, IL-6) markers were measured using a multiplex kit (TheromoFisher, Waltham, MA, USA) on a MagPix machine (Luminex; Austin, TX, USA). 

### 4.6. Immunological Profiling with Olink

OLINK technology was employed to assess the serum and urine immunological profile. The company assessed urine and serum samples (OLINK Bioscience, Uppsala, Sweden) [68,69]. Data presented as Normalized Protein Expression (NPX) values plotted against protein concentration are dimensionless, allowing for comparison of the measured protein over time but not between proteins [70]. The list of analytes was selected from a pre-existing set (Appendix A).

### 4.7. Details of Tobacco Exposure

Self-reported data were collected from the EHR with patients declaring being nonsmokers, past smokers, or current smokers during encounters with the healthcare system. Users acquire nicotine via smoking or chewing tobacco. In addition, we tested patients’ blood for cotinine using enzyme-linked immunoassay following manufacturer recommendations (Origene, Rockville, MD, USA). The cut-off point for cotinine levels was 3.75 ng/mL to decrease meaningful nicotine exposure [6,42,45,46,47,48].

### 4.8. Statistical Analysis

The Shapiro–Wilk W test and distribution plots were used to test the normality of distribution variables. Data were presented as mean (X) or median (M_e_) with variability expressed as standard deviation (SD) or interquartile ranges (IQ). Parametric variables were compared using the Welch test, while the Mann–Whitney U statistic was employed to compare non-parametric variables [71]. *d*-Cohen statistics were utilized to assess the significance of the differential [72]. Frequencies were compared using *χ*^2^. A double-sided *p*-value less than 0.05 was considered statistically significant for all tests unless specifically stated [71]. Statistical analyses were performed with Statistica 11.0 (StatSoft Inc., Tulsa, OK, USA) or SPSS 29.0 (IBM, New York, NY, USA). The figure and some statistical computations were carried out using GraphPad version 10.8 (Prism, Cambridge, MA, USA).

## Figures and Tables

**Figure 1 ijms-25-03714-f001:**
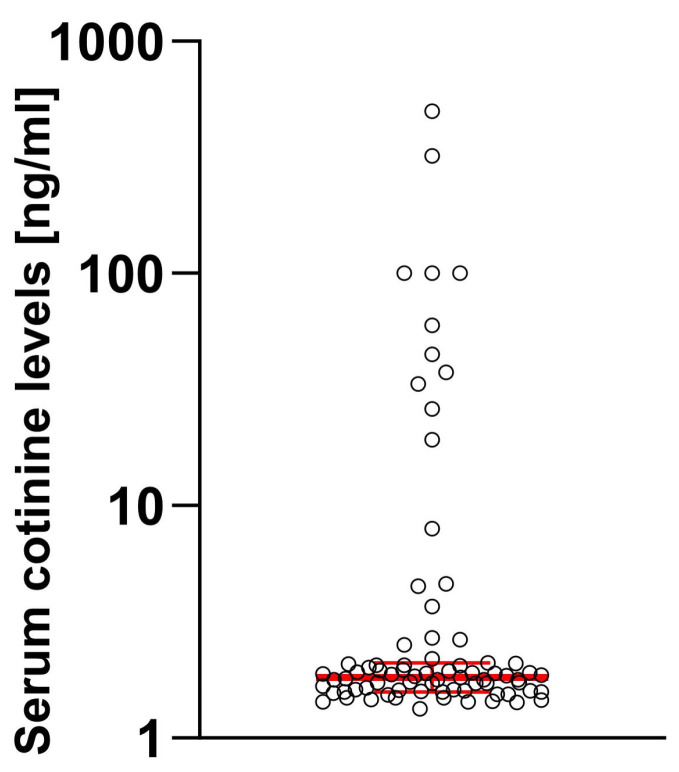
Distribution of the cotinine levels in studied samples. Log10 scale was used considering high dynamic range of observed values. Median and 95% confidence intervals are marked in red.

**Figure 2 ijms-25-03714-f002:**
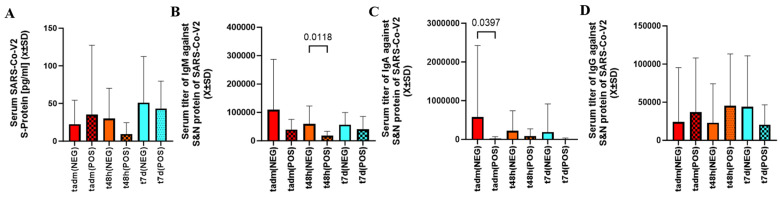
There was no difference in serum level of S-protein between patients exposed to nicotine as defined by cotinine testing (**A**). However, in patients with positive cotinine testing at admission, t_48d_ IgM level (**B**) and t_adm_ IgA levels (**C**) were significantly depressed. Smoking did not differentiate serum levels of IgG between any sampling times (**D**).

**Figure 3 ijms-25-03714-f003:**
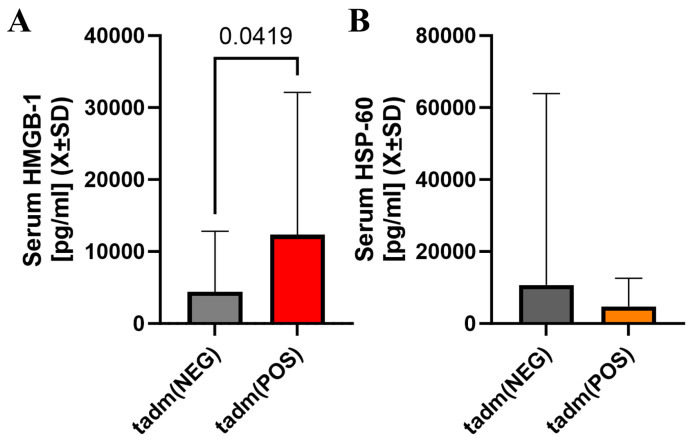
There was a statistically significant difference in serum HMGB-1 levels at admission with individuals with cotinine-positive tests having higher serum levels of HMGB-1 at admission (**A**). However, serum Hsp-60 in patients with positive cotinine testing at admission was not different vs. cotinine-negative individuals (**B**).

**Figure 4 ijms-25-03714-f004:**
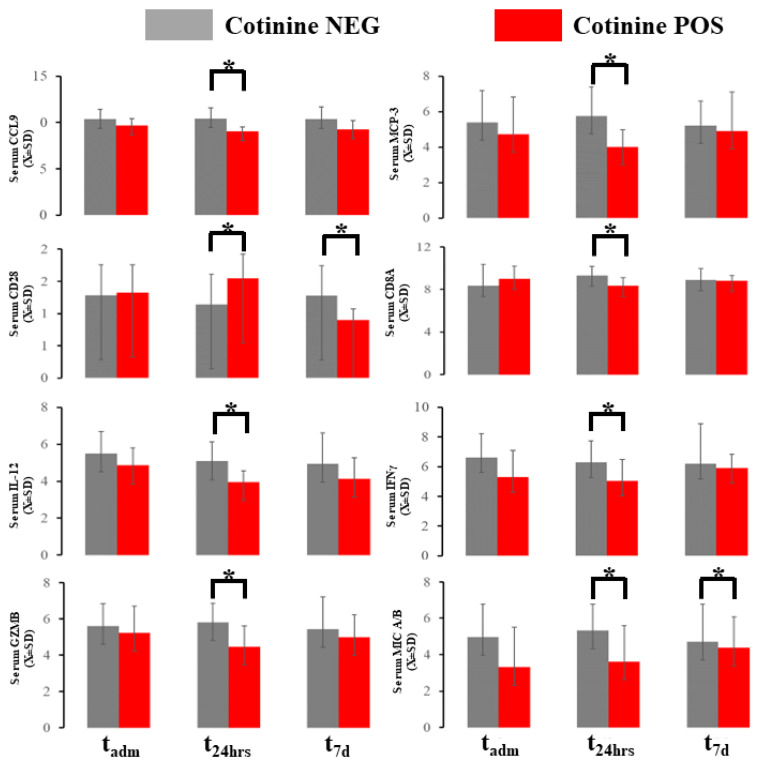
Immunological blood profiling with OLINK demonstrated several changes in the marker expressed here. * asterisk denotes a significant difference between patients testing positive vs. negative in a given time slot.

**Figure 5 ijms-25-03714-f005:**
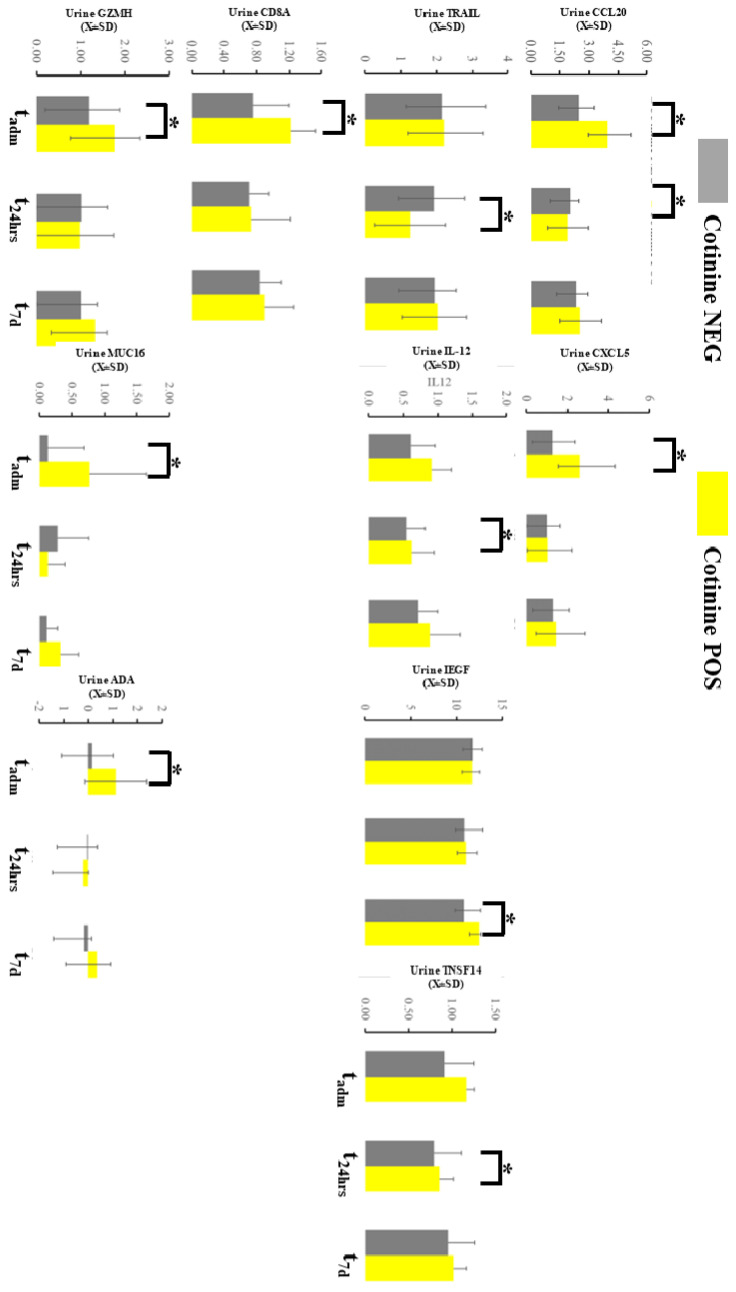
Immunological profiling of the urine with OLINK demonstrated several changes in the markers expressed here. * asterisk denotes a significant difference between patients testing positive vs negative in a given time slot.

**Table 1 ijms-25-03714-t001:** Patient demographics and clinical characteristics with comparisons between cotinine-positive versus negative individuals. (ns—non-significant), * one patient is missing demographic data.

Patients (*n* = 80)		
Demographics		Cotinine_positive_ vs. Cotinine_negative_
Age * (years) [X ± SD] *	60.4 ± 18.2	ns
Gender (female) [%] *	39.2%	ns
Race [%] *	20.3% (Caucasian) 67.1% (African American)	ns
BMI [X ± SD] *	32.1 ± 8.69	ns
**Comorbidities**		
CCI [X ± SD]	6.1 ± 2.16	ns
CVA [%]	12.5% (*n* = 10)	ns
CHF [%]	20.0% (*n* = 16)	ns
PVD [%]	10.0% (*n* = 8)	ns
COPD [%]	18.75% (*n* = 15)	ns
DM [%]	36.25% (*n* = 29)	ns
CKD [%]	25.0% (*n* = 20)	ns
ESRD [%]	2.5% (*n* = 2)	ns
**Hospital Trajectory**		
Length of Stay (days) [X + SD]	12.9 ± 15.68	ns
Admitted to the ICU (%) Duration of ICU stay (days) [X ± SD]	50.6%; 8.49 ± 18.47	ns
Mechanical ventilation (%) Duration of ICU stay (days) [X ± SD]	27.8% 5.05 ± 14.09	ns
ECMO (%) Duration of ECMO (days) [X ± SD]	3.8% 1.9 ± 12.88	ns
**Treatment**		
Remdesivir [%]	53.2%	ns
Steroids [%]	63.3%	ns
Plasma [%]	8.9%	ns
**Outcome**		
Alive at 6 months [%]	73.8%	ns

## Data Availability

The datasets used and/or analyzed during the current study are available from the corresponding authors upon reasonable request and approval from the IRB.

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
