# Peer review of "Immunological Signatures in Blood and Urine in 80 Individuals Hospitalized during the Initial Phase of COVID-19 Pandemic with Quantified Nicotine Exposure"

_ijms, 2024, doi:10.3390/ijms25073714_

Round 1

Reviewer 1 Report

Comments and Suggestions for Authors

This study implied that nicotine exposure affects biomarkers related to COVID-19. The information is very interesting and lacking in the publications. However, the characteristics of the participants were lacking. The outcomes should be more detailed in the nicotine exposure to make it more informative.

Major concerns.

1. This study focuses on nicotine from smokers.
Table 1 should include the average smoking frequency and non-smokers.

If there were an enormous difference, such as 1-10 cigarettes vs 20-30 cigarettes.
I think you can do a subgroup analysis to explore more information. (optional)

2. Did some participants receive any nicotine replacement therapy?
I think this point can be shown in the characteristics.

3. Did some participants uses e-cigarratte?
Suggest to clarify it.

Comments.

1. Line 27: "This manusript analyzes immunological...".
I suggest changing this statement because when this manuscript is accepted and published, the status of your writing will change from "manuscript" to "article".
Consider using "study" instead of manuscript.

2. Second-hand and third-hand smokers are at risk of exposure to nicotine. However, I know that it is difficult to collect the data or assess the amount and time to exposure to nicotine.
If you have an appropriate data. You can show or discuss it.

3. Line 64 suggests adding an abbreviation of the "renin-angiotensin-system" to this statement because line 80 mentioned "RAS" without a full name available in this manuscript.

4. Was this study registered in any registry?
If yes, suggest adding a registry name with an identifier of this project.

5. In Table 1, you can stratify by the age group, such as <60 vs ≥60, if your data is appropriate to subgroup.
The elderly are at high risk for the severe infection conditions. The outcome of senescence is very interesting.

6. Figure 1. It seems all plots were pooled (total) in one column.
You can rotate to verticle and add the bar with error (i.e. mean with SD or geometric mean with 95%CI)

(Optional) If you want to make it more informative.  I suggest a subgroup depending on your assumption with statistical comparison, with a pooled (total) on the right side column.

Typos.

1. Please check the last column of the Table 1.

Author Response

Paper was significantly modified. We clarify number of patients enrolled in the study. Additional statistical were calculated. We added one main text figure and one supplemental figure. We add additional statistical results and statistics. The manuscript was revised for the purpose of increase readability and the form.

All these changes necessitated addition of the additional author. An adequate authorship change form has been added.

Reviewer #1

We truly appreciate this input. A lot of suggestions brought up by this reviewer were on our minds but the numerical size of the group prevented us from executing analysis. We appreciate the remarks. We incorporate a lot of them into discussion.

  1. This study focuses on nicotine from smokers. Table 1 should include the average smoking frequency and non-smokers.
    1. We could have used this data for visualization, but there was no difference in the frequency/values of the demographic variables. We added an appropriate remark in the Table.
  2. If there were an enormous difference, such as 1-10 cigarettes vs 20-30 cigarettes.
    I think you can do a subgroup analysis to explore more information. (optional)
    1. The differences were significant. They reflect smoking habits. This means that we had to clump all smokers in one group. When we tried to stratify smokers into “usage” levels, we found that we have three individuals (n=3) in very heavy use and some in little use. Ultimately, we were unsure if smoking has a linear or threshold effect on immunological response.
  3. Did some participants receive any nicotine replacement therapy?
    Though this question is asked regularly, all our responders said no. We tried to see if the nicotine patch was prescribed to them, but during the COVID-19 pandemic, nicotine patches were not utilized for fear of worsening patients' conditions. We would love to implement this suggestion but a larger study is needed.
  4. Did some participants usee-cigarette?
    This question is part of intake but every participant answered no. We think this is a biased response. We added a comment about using vaping/-e-cigarettes in to the discussion section.
  5. Line 27: "This manuscript analyzes immunological...".
    1. Changed
  6. Second-hand and third-hand smokers are at risk of exposure to nicotine. However, I know that it is difficult to collect the data or assess the amount and time to exposure to nicotine. If you have an appropriate data. You can show or discuss it.
    1. We do not have this data but we made a point in the discussion. Ultimately, second or first hand smoking are equally harmful despite the intent.
  7. Line 64 suggests adding an abbreviation of the "renin-angiotensin-system" to this statement because line 80 mentioned "RAS" without a full name available in this manuscript.
    1. Added
  8. Was this study registered in any registry? If yes, suggest adding a registry name with an identifier of this project.
    1. The study is not registered.
  9. In Table 1, you can stratify by the age group, such as <60 vs ≥60, if your data is appropriate to subgroup. The elderly are at high risk for the severe infection conditions. The outcome of senescence is very interesting.
    1. We did an analysis but no relationship was identified.
  10. Figure 1. It seems all plots were pooled (total) in one column. You can rotate to verticle and add the bar with error (i.e. mean with SD or geometric mean with 95%CI)
    1. So this figure is on a logarithmic scale to show very dynamic range of the cotinine level distribution. Data are non-parametric so we used non-parametric descriptive statistics. We tried to do a subgroup, but there were not enough subjects per group to have a meaningful analysis.
  11. (Optional) If you want to make it more informative.  I suggest a subgroup depending on your assumption with statistical comparison, with a pooled (total) on the right side column.
    1. Review Table 1
    1. Proofed

Reviewer 2 Report

Comments and Suggestions for Authors

This research studied the nicotine exposure effect on SARS-CoV-2 infection. The background was written well, and authors carefully designed the research method. Real-world data is another great advantage.

However, several issues need to be fixed before publication. 

1. Please consider update to a concise title. It's too wordy now. 

2. The authors only reported the percentage of exposed group, please be clear about the N, too.

3. In Table 1, I'd like to see the sample numbers and baseline status for each group separately. 

4. Figure 2 needs to be re-arranged just like Figure 3 and 4. Please put the bars of same timepoint together for comparison. Also, Figure 3 and 4 seems need some format adjustment to correct the overlapped parts and make sure x-axis legend is showing properly. 

Author Response

Reviewer #2

  1. Please check the last column of the Table 1.
    1. We recheck the Table 1. Table was also re-formatted
  2. Please consider updating to a concise title. It's too wordy now. 
    1. Adjusted in title
  3. The authors only reported the percentage of exposed group, please be clear about the N, too.
    1. We added this information
  4. In Table 1, I'd like to see the sample numbers and baseline status for each group separately. 
    1. There is no difference In the demographic as specified in the statistical analysis. We can provide this data but it will be redundant at this point
  5. Figure 2 needs to be re-arranged just like Figure 3 and 4. Please put the bars of same timepoint together for comparison. Also, Figure 3 and 4 seems need some format adjustment to correct the overlapped parts and make sure x-axis legend is showing properly. 
    1. Figures were redone.
